# [RE] Reproducibility Study of "Explaining Temporal Graph Models Through an Explorer-Navigator Framework"

**Andreas Berentzen**                                    *andreas.berentzen@student.uva.nl*
*University of Amsterdam*

**Helia Ghasemi**                                        *helia.ghasemi@student.uva.nl*
*University of Amsterdam*

**Christina Isaicu**                                     *christina.isaicu@student.uva.nl*
*University of Amsterdam*

**Jesse Wonnink**                                        *jesse.wonnink@student.uva.nl*
*University of Amsterdam*

## Abstract

This paper seeks to reproduce and extend the results of the paper "Explaining Temporal Graph Models Through an Explorer-Navigator Framework" by Xia et al. (2023). The main contribution of the original authors is a novel explainer for temporal graph networks, the Temporal GNN Explainer (T-GNNExplainer), which finds a subset of preceding events that "explain" a prediction made by a temporal graph model. The explorer is tested on two temporal graph models that are trained on two real-world and two synthetic datasets. The explorer is evaluated using a newly proposed metric for explanatory graph models. The authors compare the performance of their explorer to three baseline explainer methods, either adapted from a GNN explainer or developed by the authors. The authors claim that T-GNNExplainer achieves superior performance compared to the baselines when evaluated with their proposed metric. This work reproduces the original experiments by using the code (with minor adjustments), model specifications, and hyperparameters provided by the original authors. To evaluate the robustness of these claims, the method was extended to one new dataset (MOOC). Results show that the T-GNNExplainer performs best on most, but not all metrics reported in the original findings. We conclude that the main lines of this paper hold up even though all results are less pronounced than claimed. Results show that the T-GNNExplainer does not perform similarly across different T-GNN models, precise dataset specifications are needed to obtain high performance, and there are simpler, less computationally costly explainer methods that could offer competitive results.

## 1    Introduction

Dynamic graph-structured data can be seen as sequences of events between nodes that happen over time, and exist in many applications such as social networks (Pereira et al., 2018)(Gelardi et al., 2021). Temporal Graph Neural Networks (T-GNNs) learn representations of these structures and make predictions on future events (Xia et al., 2023). The rationale for how graph models arrive at these predictions is difficult to interpret (Luo et al., 2020), and this lack of human-intelligible explanations means these powerful tools lack the transparency crucial in establishing fairness, safety, and trust in their output (Doshi-Velez & Kim, 2017). The opacity of graph models could hinder the possibility to ensuring ethical outcomes if potential biases or discrimination arise from their predictions, especially in high-stakes domains like healthcare (Burrell, 2016).

Graph explainers identify important subgraphs of nodes and events that were instrumental in a model's prediction for a target event (Luo et al., 2020). Most methods for explaining these predictions exist for Graph

Neural Networks (GNNs), which focus on static graphs (GNNExplainer (Ying et al., 2019), PGExplainer (Luo et al., 2020) and Sub- graphX (Yuan et al., 2020)). These methods cannot be directly applied to explaining T-GNNs, since the time-varying structures of dynamic graphs are not captured by these explainers (He et al., 2022).

The authors of this paper propose the T-GNNExplainer, which provides an instance-level search-based model-agnostic post-hoc explanation for predictions made by temporal graph models. "**Instance-level**" means an explanation is provided only for one prediction instead of at a global level, "**search-based**" means the method explores subsets of possible solutions, "**model-agnostic**" means the explainer should be able to explain any temporal graph model, and "**post-hoc**" means the explanation is based on the output of the model without direct access to the actual steps that occurred during model training.

This paper is structured as follows: In **Section 2**, we explain the scope of reproducibility, summarizing the authors' main claims and the experiments we ran to verify their claims. In **Section 3**, we provide the reader with a summary of the model proposed in the original paper and the experimental methodology used to run the experiments. In **Section 4** we describe all the datasets and in **Section 5** we explained our use of the hyperparameters. In **Section 6** we described the experimental setup and code, where we explain the novel dataset and the computational requirements. **Section 7** provides the replicated results, comparison to the original results, and results that extend beyond the original paper. We end with a discussion in **Section 8**. Beyond the references we have also have an appendix, A to B, that provides further information on the specifics of the experimental setup for those looking to implement this reproducibility study, further metrics, results, and suggestions for further research.

## 2   Scope of Reproducibility

The main contribution of the authors is a novel explainer for T-GNNs. They claim superior performance of their explainer by evaluating performance in comparison to other baselines explainers, evaluating the efficiency of their method, and the conciseness of the output compared to the other explainers.

The first metric they use to evaluate performance is called the Area under the Fidelity-Sparsity Curve (AUFSC). It is composed of two metrics (fidelity and sparsity), each of which is used to evaluate graph explainer models (Amara et al., 2022)(Agarwal et al., 2023)(Liu et al., 2021). The curve expresses the relationship between the fidelity (i.e. how accurately the T-GNN makes a similar prediction when it takes as input only the nodes it selects as explanatory) and sparsity (i.e. how few nodes are used to give the explanation). The area under the curve is a measure of the commutative fidelity over a set of sparsity threshold intervals. High fidelity and high sparsity result in a higher AUFSC which indicates better performance. The second metric, Best Fidelity, simply measures the highest fidelity achieved by an explainer without the sparsity limitation, averaged over all test data. The metrics are explained in further detail in Section 3.10.

The main claims that are investigated in the current paper are:

1. Compared to the other baselines, T-GNNExplainer surpasses performance in the **AUFSC** and **Best Fid** metrics by up to ~50%.

2. The T-GNNExplainer is model agnostic in regards to the underlying predictor model.

The aim of this paper is to verify the original authors' main claim by reproducing their main findings, and testing the robustness of this claim by extending their results to a new and different kind of dataset. We perform further analysis on the explanatory graphs and the interaction between the T-GNNExplainer and the underlying model. Further goals of the paper are to increase accessibility and reproducibility of this work by providing supplementary instructions for running the code and in-depth explanations for understanding the methodology.

There are other claims made in the paper in regards to the conciseness of the generated graph and the navigators efficiency. We do not investigate these claims due to computation and time cost.

# 3  Methodology

The code is publicly available in a zip file of supplementary materials on the OpenReview page for this paper. The data is not included in the zip file, but the top-level ReadMe in the code provides instructions for downloading the data. There is some inconsistency between the nomenclature used in the paper and the corresponding terminology in the code. For the purpose of this work, we follow the terminology used in the paper.

## 3.1  Notation

The notation used in the paper is as follows: $\mathcal{G} = (\mathcal{N}, \mathcal{S})$ is a temporal graph. $\mathcal{N}$ is a list of the nodes, and $\mathcal{S} = \{e_1, e_2, \dots\}$ sequence of timestamped events. Each event $e_i = \{n_{u_i}, n_{v_i}, t_i, \text{att}_i\}$ has a timestamp $t_i$ and occurs from node $n_{u_i}$ to node $n_{v_i}$ with the attribute $\text{att}_i$, which is a list of indefinite even length filled with feature values. $\mathcal{G}^k$ is the graph at timestamp $k$ and $\mathcal{R}^k$ is the subset of events in graph $\mathcal{G}^k$ found by the explainer. $N_r$ is a hyperparameter controlling the amount of nodes that the explanatory subgraph is allowed to have.

The function $f(\cdot)$ represents a trained temporal graph model. $f(\mathcal{G}^k)[e_k]$ is the probability prediction of the event $e_k$ that occurs when the graph $\mathcal{G}^k$ is input into the network. The output is a logit value. $Y_k$ is a value of either 0 or 1 that classifies the prediction of the network for whether or not an event occurs at the time step $k$.

## 3.2  T-GNNExplainer

We explain the Explorer-Navigator framework in detail to give a more intuitive understanding of the model intended functioning.

A temporal graph network predicts whether a target event $e_k$ will occur or not (Rossi et al., 2020)(Xu et al., 2020). The purpose of the T-GNNExplainer is to find a set of preceding events that explain why the model made this prediction.

This subset of events $\mathcal{R}^k$ henceforth referred to as the "explanatory subgraph" is found from a set of candidate events. These candidate events are selected through three hyperparameters. The first hyperparameter ($\boxed{\text{n\_hops}}$ in the code) limits them spatially to be within a k-hop neighborhood of the target event (i.e., 2 hops). The second ($\boxed{\text{num\_neighbors}}$ in the code) restricts them temporally to the target event (i.e., within the 10 most recent events). $\mathcal{R}^k$ finally, is further limited by the hyperparameter $N_r$, which constrains the maximum number of nodes that the explanatory subgraph can have. In the code, this number is hard coded to 20. The reasoning is that if the subgraph is too large e.g. if $|\mathcal{R}^k| = |\mathcal{G}^k|$, then the explanatory power of $\mathcal{R}^k$ becomes tautological, losing usefulness in its lack of specificity.

The explainer module is made up of two parts, the navigator and the explorer, that work together to "prune" a graph of unimportant events related to the target event $e_k$, resulting in a graph of only the most relevant events that lead to the prediction of the target event.

## 3.3  Navigator

The navigator is inspired by the "explanation network for node classification" from the parameterized explainer proposed by Luo et al. (2020). The navigator $h_\theta(e_j, e_k)$ proposed in this paper is a two-layer multi-layer perceptron (MLP) that learns a correlation between two events $e_j$ (the candidate) and $e_k$ (the target).

To infer the relationship between two specific events, the features of both events are concatenated into a vector and input into the network:

$$Z_{e_j, e_k} = [X_{n_{u_k}} \| X_{n_{v_k}} \| \text{Time}(t_k) \| \text{att}_k \| X_{n_{u_j}} \| X_{n_{v_j}} \| \text{Time}(t_k) \| \text{att}_k]^T$$

$X$ represents the node feature matrix, $Time(\cdot)$ is a harmonic function that encodes the real-valued timestamp into a learnable vector adapted from (Xu et al., 2020), and || represents concatenation.

Given that the navigator is an MLP that takes as input the concatenated features of two events to learn a relationship between them, this indicates that the network learns similarities between events. So when the explorer (explained in section 3.4 below) uses the navigator to remove "unimportant" events from the search space, the navigator selects dissimilar events to the target event.

### 3.4   Explorer

The explorer uses a modification of Monte Carlo Tree Search (MCTS). The explorer searches a tree of nodes, where each child node is a subgraph of its parent node, which has one fewer event than the parent before it.

The root node is initialized as a subset of $\mathcal{G}^k$ that satisfies the constraints imposed by temporal and spatial hyperparameters.

Starting from the root node, the explorer selects and expands child nodes. It is important to note that these child nodes are not stored in memory. Rather, the explorer keeps track of a list of previously expanded events at each node $\mathcal{C}(\mathcal{N}^i)$. This entire process is conducted through multiple rounds (i.e. rollouts), a hyperparameter that is set to 500 in this paper.

Each rollout starts with the root node and we prune events until we reach a leaf node, which is a child that satisfies the hyperparameter criteria that it has fewer than 5 events. In the code itself, a leaf node is defined as a subgraph with 1 event in the code, and less than 5 events in the paper.

Child nodes are selected and expanded until a leaf is reached. The search tree of all nodes is not not stored in memory across rollouts, rather each rollout starts with the root and a list of events that were expected and expanded in previous rollouts.

### 3.5   Node Selection

The search path is selected using the following formula:

$$e^* = \underset{e_j \in \mathcal{C}(\mathcal{N}^i)}{\arg\max} \left( \frac{c(\mathcal{N}^i, e_j)}{n(\mathcal{N}^i, e_j)} + \lambda \frac{\sqrt{\sum_{e_l \in \mathcal{C}(\mathcal{N}^i)} n(\mathcal{N}^i, e_l)}}{1 + n(\mathcal{N}^i, e_j)} \right) \tag{1}$$

Since the nodes themselves are not stored in memory, a child node is "selected" by removing an optimal event e* from the current node $N^i$. This equation says that from events that have already been expanded, select an event that has a high average reward value from previous rounds (exploitation term) but has not been explored as much (exploration term). The $\lambda$ is a hyperparameter set by the paper authors that controls how much exploration is encouraged.

### 3.6   Node Expansion

Instead of generating and searching a tree of all possible subgraphs, the explorer uses the navigator to create subgraphs that are relevant to its search. The navigator finds the least important event in relation to the target/predicted event $e_k$:

$$e^* = \underset{e_j \in \mathcal{N}^i / \mathcal{C}(\mathcal{N}^i)}{\arg\min} h_\theta(e_j, e_k) \tag{2}$$

The explorer then creates a new child node by removing $e^*$ from the current node $N_i$. It is important to note that the navigator only searches events that were not already removed in previous rollouts, so that we do not end up removing the same event in every rollout.

### 3.7 Reward & Backpropagation

Child nodes are selected and expanded until we reach a leaf node. Once we have reached this leaf node, a reward is calculated through the negative cross-entropy loss based on the work of Farnia & Tse (2017):

$$\min_{\mathcal{R}^k} - \sum_{c=0,1} \mathbb{1}(Y_k = c) \log P(Y_{new} = c | \mathcal{R}^k) \tag{3}$$

The loss looks at how much the probability of a prediction ($Y_k$) of the original graph ($G_k$) changes when the input is limited to a specific subgraph ($\mathcal{R}^k$). If the probability changes a lot, then that subgraph is not a good explanation for the prediction made on event $e_k$ by the original graph. The outcome of the loss function is seen as the reward for the back-propagation. The back-propagation step itself, consists of adding the cumulative reward value for each subgraph/node $c(\mathcal{N}^i, e_j)$, and how many times it's received a reward $c(\mathcal{N}^i, e_j)$.

### 3.8 Selecting the explanatory subgraph

The explorer-navigator process is repeated through several rollouts to explore different possible subgraphs. The best explanation for the prediction $e_k$ is the subgraph that has the highest cumulative reward and has the simplest explanation according to the sparsity threshold ($|\mathcal{R}^k| \leq N_r$ in equation 4).

### 3.9 Optimal explainer

The optimal explainer $g^*$ is the minimum cross-entropy averaged over all predictions for k target events

$$g^* = \arg\min_g -\frac{1}{K} \sum_{k=1}^{K} [\mathbb{1}(Y_k = 1) \log \sigma(f(\mathcal{R}^k)[e_k]) + \mathbb{1}(Y_k = 0) \log(1 - \sigma(f(\mathcal{R}^k)[e_k]))] \tag{4}$$

$$\text{Subject to } \mathcal{R}^k = g(e_k, \mathcal{G}^k, f(\cdot)) \text{ and } \mathcal{R}^k \subseteq \mathcal{G}^k \text{ and } |\mathcal{R}^k| \leq N_r$$

where $N_r$ is the hyperparameter that controls the size of $\mathcal{R}^k$.

### 3.10 Evaluation

The two metrics used to evaluate the performance of the explainers are the Area Under the Fidelity-Sparsity Curve (**AUFSC**), and **Best Fid** (best fidelity). The formula for sparsity is given by: $Sp = |\mathcal{R}^k|/|\mathcal{G}^k|$ Sparsity is a ratio between the size of the explanatory subgraph and the input graph $\mathcal{G}^k$.

The formula for Fidelity is given by:

$$Fid(f(\mathcal{G}^k)[e_k], f(\mathcal{R}^k)[e_k]) = \mathbb{1}(Y_k = 1)(f(\mathcal{R}^k)[e_k] - f(\mathcal{G}^k)[e_k]) + \mathbb{1}(Y_k = 0)(f(\mathcal{G}^k)[e_k] - f(\mathcal{R}^k)[e_k]) \tag{5}$$

Fidelity measures the difference between the logit probability output by the temporal network $f(\cdot)$ when the input is the original graph $\mathcal{G}^k$ and when the input is the explanatory subgraph $\mathcal{R}^k$. a logit score of $+1$ implies that the explanatory subgraph is a better prediction than the original graph. The model aims to maximize the fidelity.

The **AUFSC** is a numerical score that measures the area under the graph for the fidelity values at sparsity intervals between 0 and 1 ($|\mathcal{R}^k| = |\mathcal{G}^k|$). A higher value for AUFSC implies the explorer achieves high fidelity, even at a low sparsity threshold.

The goal is to have high fidelity, which means that $f(\mathcal{R}^k)[e_k]$ produces a logit probability that ideally surpasses the original prediction $f(\mathcal{G}^k)[e_k]$, while keeping the explanatory subgraph $\mathcal{R}^k$ as simple as possible. The purpose is not to achieve the same probability output on $f(\mathcal{R}^k)[e_k]$ as $f(\mathcal{G}^k)[e_k]$ (fidelity = 0) but rather to obtain a subset that increases the logit number for this prediction as much as possible (positive fidelity). In a way, we are picking the best events to generate this prediction which we call the "explanatory subgraph".

The metric **Best Fid** simply measures highest fidelity achieved by an explainer, ignoring sparsity altogether. The only limitation is the hyperparameter $N_r$ controlling the maximum amount of nodes allowed.

### 3.11 Baselines - Other Explainers

The authors compare the performance of the T-GNNExplainer with three baseline explainer methods: PG-Explainer (Luo et al., 2020), ATTN and PBONE (Xia et al., 2023). In contrast to T-GNNExplainer, the baselines are not search-based methods. The original PGExplainer is a parametrized explainer for GNNs that provides a model-level explanation over multiple event predictions. The authors adapt PGExplainer to output explanations for temporal graph structures at an instance-level. The two other explainers were modelled by the authors themselves. Firstly, there is ATTN which is an attention-based explainer that averages the weight values over all attention layers of either TGAT or TGN. PBONE generates an explanation by removing or "perturbing" one candidate event at a time to compute its importance in a prediction (Xia et al., 2023).

### 3.12 Target models

The authors test the explainer on two target models, namely TGN and TGAT. TGAT is a temporal graph attention layer that is used in neural networks. This technique uses self-attention and stacks TGAT layers to recognize the node embeddings as functions of time (Xu et al., 2020). Temporal Graph Networks (TGN) is a deep learning framework where dynamic graphs are represented as sequences of timed events (Rossi et al., 2020). TGN consists of a combination of memory modules and graph-based operators. In contrast to TGAT, TGN uses Multi-head Attention in their network.

## 4 Datasets

The dataset collection consists of bipartite graphs and unipartite graphs, all of which are directed. Bipartite graphs consist of two types of nodes (e.g. users and pages). Events in the datasets describe one type of node interacting with the other type (Chang & Tang, 2014). Unipartite graphs consist of one type of node, among which events occur (Chang & Tang, 2014).

The original paper experiments with two classes of datasets with very distinct characteristics. One is a bipartite graph network describing an online collaboration process with extensive feature information, and the other is a dataset generated by a statistical process in a small-scale graph with predefined, static event relationships.

### 4.1 Data format

The original paper makes some claims about the data structures that conflict with the implementation of the code. This section highlights implicit assumptions and discrepancies relevant to our discussion in Section 8 about the precise data specifications needed to run the models.

It is first important to note that every dataset used is a directed graph, the format consists of events that occur from node $n_{u_i}$ to node $n_{v_i}$. There can be multiple events from node $n_{u_i}$ to node $n_{v_i}$ (i.e. a user updates a page multiple times). The original paper states that there could be deletion or internal update events $e_i = \{n_{u_i}, null, t_i, \text{att}_i\}$, however, the implementation for the T-GNNExplainer does not allow for null events, therefore there are no self-updates or self-deletions of nodes.

Even though the original paper claims that the T-GNNExplainer takes into account the continuous-time dynamic properties of the data (Xia et al., 2023), only the underlying predictor models do this. Analysis of the methodology and code reveals that T-GNNExplainer only looks at events ordinally, and ignores the amount of time that passes between events.

### 4.2 Synthetic datasets

The synthetic datasets are based on a unipartite temporal graph that contains 4 nodes, and 4 possible edge-events. The possible events within this graph adhere to predefined excitatory and inhibitory relations. These predefined relations are used in a multivariate Hawkes process to generate a dataset of timestamped events. (Hawkes, 1971) implemented in the Tick Python library (Bacry et al., 2018). A random process is

used to generate noise in the form of false events. In summary, the synthetic datasets are relatively small datasets that describe the occurrence of four unique events over time.

### 4.3 Real-World datasets

The two real-world datasets used in the original paper, Wikipedia and Reddit (Kumar et al., 2019), are bipartite graphs that describe editing processes between users and web pages. In both datasets, this process follows a roughly 1:9 ratio between web pages and users. Each event is accompanied by a 172 numeric features vector that describes the editing event.

It should be noted that although it is claimed these datasets are bipartite, about 10% of the connections involve nodes that are contained in both sets, this would disprove the bipartite characteristic. We have not found a reason for this phenomenon in the original paper or from the authors who compiled the datasets (Kumar et al., 2019).

### 4.4 New dataset

To validate the robustness of the claims made by the original authors regarding AUFSC performance, a new dataset is included in the experimentation. **MOOC** is a network representing student interaction from online course content units (Kumar et al., 2019).

**MOOC** exists in a desirable interposition of dataset characteristics to the original datasets. It is equal in size but has fewer features and can serve to test the generalizability of the proposed methodology on real-world temporal graph processes with limited event information.

Table 1: Dataset attributes. Where # represents "Number", so "Number of Events" and "Number of Features".

| Name | # of Events | # of Features | Type | Time period | Time granularity |
|------|-------------|---------------|------|-------------|------------------|
| Wikipedia | 157,474 | 172 | bipartite | 1 month | seconds |
| Reddit | 672,447 | 172 | bipartite | 1 month | seconds |
| MOOC | 411,749 | 4 | bipartite | ~298 days | seconds |
| V1 | 10,000 | 0 | unipartite | ~5000 seconds | <seconds |
| V2 | 10,000 | 0 | unipartite | ~5000 seconds | <seconds |

All data sets have a 70% / 15% / 15% train / validation / testing scheme based on timestamps for both TGAT and TGN. The most recent 30% is used for the validation and test sets (Kumar et al., 2018)(Kumar et al., 2019).

## 5 Hyperparameters

The experimental setup is copied from Section 5.2 and the Appendix of the original paper (see Appendix B below for further details).

For the novel dataset, minor changes have been made. Firstly, before training the graph prediction models (TGAT and TGN) the embedding dimensionalities are adjusted to match the dimensionality of the feature information in the novel dataset.

The experiments were run on two seeds; the original seed 2020, and a new seed 2. See the discussion section for further comments about the incomplete seed set-up in the original code.

Due to the limitation of computational resources, we chose to focus our hyperparameter tuning efforts on recreating the hyperparameter analysis performed by the original authors in their Appendix Section A.7. Our experiment is performed on the TGN model trained on the Wikipedia dataset, where $\lambda$, the navigator's exploration parameter, is set to 1, 5, 10, and 100.

Table 2: Other hyperparameters from the paper and justifications regarding why they were excluded from further studies.

| Hyperparameters omitted from tuning | | |
|---|---|---|
| **Hyperparameters** | **Value** | **Explanation for omission** |
| Explanation level | Event | Graph-level explanations are not implemented by the authors but there is evidence in the code of their intention to implement this option. |
| Rollouts | 500 | The authors performed rollout analysis in the Appendix, but we faced compute limitations |
| $N_r$ (See Section 3.9) | 20 | This hyperparameter was hardcoded by the original authors, code refactoring was out of scope for this paper. |
| Learning rate | $1e^{-4}$ | Limitation of computational resources. |
| Time dim | 172 or 4 | Values are very specific to the dataset. |
| Node Feature Dimension | 172 or 4 | Values are very specific to the dataset. |
| Edge Feature Dimension | 172 or 4 | Values are very specific to the dataset. |
| Training epoch | 10 or 100 | Limitation of computational resources. |

## 6 Experimental setup & Code

The following experiments were performed: 1) Reproduction on the original performance claims across three seeds, 2) Extension of performance evaluation to one new datasets, 3) hyperparameter tuning of $\lambda$, the value responsible for the "exploration" of the navigator.

The repository can be found here https://github.com/cisaic/tgnnexplainer.

### 6.1 Reproduction

To reproduce the original results, we train the models (TGAT, TGN) on the original datasets (Wikipedia, Reddit, Synthetic V1, Synthetic V2). We then train the explainers (T-GNNExplainer, PGExplainer, ATTN, PBONE) on the trained temporal graph networks from the previous step. Finally, the performance is evaluated using the evaluation methods in the authors' original code to measure AUFSC and Best Fid scores (evaluation detailed in section 3.10.

### 6.2 Extension with novel datasets

New datasets need to adhere to a number of constraints to train explainer models, which include data format constraints and content constraints. Before the new dataset can be used in training, it needs to be processed to match the navigator feature vector as described in Section 4.4 The full experimental setup of the new datasets is described in Appendix B.

### 6.3 Computational requirements

For this project, we ran an initial set of experiments (not included in the paper due to lack of seeding) on a cluster with each node having 24 CPU cores (AMD EPYC 7F72) and one A100 GPU with 40GB RAM.

The updated experiments, which were run over 2 seeds and include hyperparameter tuning, were run on nodes with 4x T4 GPUs with 16GB RAM each and 64 CPUs. We ran each experiment with 8-64 CPU cores and one GPU, since the T-GNNExplainer can benefit from CPU parallelism.

Training times were also tracked:

In our initial set of experiments, the average training time was 4 hours for each model, and 26 hours for each explainer. The total number of computational hours spent were 88 hours for the training of the models and 345 hours for the running the explainers (including re-runs for failed runs).

For the updated experiments, the average training time was 7 hours for each model per seed, 53 hours for the T-GNNExplainer on a single CPU, and 12 hours for all other explainers combined on a single CPU (per model). The computational hours required for the results presented in this research can be split up into 180 hours for training of the models, and 165 hours for running the explainers across 8 CPUs (including re-runs for failed runs). The cost of the updated experiments totaled over €900.

We discovered that GPU resources were not optimally allocated in sections of the authors' original code, which resulted in slower computation that was performed on CPUs. We made minor adjustments to improve efficiency but did not perform a thorough analysis and refactoring of the code.

## 7 Results

### 7.1 Results on original paper

In this section, we report our reproduced results in Table 3, Table 4, Table 5 and Table 6 for the real-world and synthetic datasets for all target models. New results are compared to the old results from the original paper, and Delta refers to the difference between the two. Bold values highlight the explainer with the highest value for a given metric. As far as we know, the original papers results were not run over multiple seeds.

Table 3: Best fidelity and AUFSC achieved by each explainer on Wikipedia dataset, averaged over two seeds.

| | | Wikipedia | | | | | | | | | | |
|---|---|---|---|---|---|---|---|---|---|---|---|---|
| | | TGAT | | | | | | TGN | | | | |
| | | BestFid | | | AUFSC | | | BestFid | | | AUFSC | |
| | Old | New | Delta | Old | New | Delta | Old | New | Delta | Old | New | Delta |
| **ATTN** | 0.891 | 0.461 | -0.430 | 0.564 | 0.510 | -0.054 | 0.479 | 0.034 | -0.445 | 0.073 | 0.047 | -0.026 |
| **PBONE** | 0.027 | 0.998 | 0.971 | -2.227 | 1.035 | 3.262 | 0.296 | 0.111 | -0.185 | -0.601 | 0.127 | 0.728 |
| **PG** | 1.354 | -0.475 | -1.829 | 0.692 | -0.420 | -1.112 | 0.464 | -0.532 | -0.996 | -0.231 | -0.504 | -0.273 |
| **T-GNN** | **1.836** | **1.236** | -0.600 | **1.477** | **1.390** | -0.087 | **0.866** | **0.413** | -0.453 | **0.590** | **0.496** | -0.094 |

Table 4: Best fidelity and AUFSC achieved by each explainer on Reddit dataset, averaged over two seeds.

| | | Reddit | | | | | | | | | | |
|---|---|---|---|---|---|---|---|---|---|---|---|---|
| | | TGAT | | | | | | TGN | | | | |
| | | BestFid | | | AUFSC | | | BestFid | | | AUFSC | |
| | Old | New | Delta | Old | New | Delta | Old | New | Delta | Old | New | Delta |
| **ATTN** | 0.658 | -0.980 | -1.638 | -0.654 | -0.889 | -0.235 | 0.575 | -2.223 | -2.798 | 0.289 | -2.186 | -2.475 |
| **PBONE** | 0.167 | **0.725** | 0.558 | -2.492 | **0.798** | 3.290 | 0.340 | **-0.876** | -1.216 | -0.256 | **-0.807** | -0.551 |
| **PG** | 0.804 | -0.877 | -1.681 | -0.369 | -0.796 | -0.427 | 0.679 | -2.438 | -3.117 | 0.020 | -2.418 | -2.438 |
| **T-GNN** | **1.518** | 0.561 | -0.957 | **1.076** | 0.753 | -0.323 | **1.362** | -1.570 | -2.932 | **1.113** | -1.496 | -2.609 |

Performance on the Reddit dataset is an outlier compared to other datasets. The PBONE method performs comparable to the T-GNNExplainer on both models trained on this dataset. All explainers have lower performance on this dataset for both models (i.e. the delta values are mostly negative), except for PBONE on TGAT.

Table 5: Best fidelity and AUFSC achieved by each explainer on Synthetic v1 dataset, averaged over two seeds.

| | **Synthetic v1** | | | | | | | | | | | |
| | **TGAT** | | | | | | **TGN** | | | | | |
| | | BestFid | | | AUFSC | | | BestFid | | | AUFSC | |
| | Old | New | Delta | Old | New | Delta | Old | New | Delta | Old | New | Delta |
|---|---|---|---|---|---|---|---|---|---|---|---|---|
| **ATTN** | 0.555 | 1.136 | 0.581 | 0.390 | 1.166 | 0.776 | 2.178 | 0.452 | -1.726 | 1.624 | 0.477 | -1.147 |
| **PBONE** | 0.044 | 1.431 | 1.387 | -2.882 | 1.447 | 4.329 | 0.000 | 0.693 | 0.693 | -3.311 | 0.721 | 4.032 |
| **PG** | 0.476 | 0.634 | 0.158 | -0.081 | 0.676 | 0.757 | 2.006 | -0.404 | -2.410 | 0.626 | -0.337 | -0.963 |
| **T-GNN** | **0.780** | **1.472** | 0.692 | **0.666** | **1.596** | 0.930 | **2.708** | **0.828** | -1.880 | **2.281** | **0.945** | -1.336 |

Table 6: Best fidelity and AUFSC achieved by each explainer on Synthetic v2 dataset, averaged over two seeds.

| | **Synthetic v2** | | | | | | | | | | | |
| | **TGAT** | | | | | | **TGN** | | | | | |
| | | BestFid | | | AUFSC | | | BestFid | | | AUFSC | |
| | Old | New | Delta | Old | New | Delta | Old | New | Delta | Old | New | Delta |
|---|---|---|---|---|---|---|---|---|---|---|---|---|
| **ATTN** | 0.605 | 0.859 | 0.254 | 0.291 | 0.874 | 0.583 | 0.988 | 0.217 | -0.771 | -0.634 | 0.237 | 0.871 |
| **PBONE** | 0.096 | 1.856 | 1.760 | -4.771 | 1.901 | 6.672 | 0.320 | 0.811 | 0.491 | -5.413 | 0.860 | 6.273 |
| **PG** | 1.329 | -0.664 | -1.993 | -0.926 | -0.641 | 0.285 | 1.012 | -0.323 | -1.335 | -1.338 | -0.328 | 1.010 |
| **T-GNN** | **1.630** | **2.286** | 0.656 | **1.331** | **2.438** | 1.107 | **4.356** | **1.001** | -3.355 | **3.224** | **1.136** | -2.088 |

In our reproduction of the experimentation, T-GNNExplainer largely does not achieve the same performance as in the original experimentation in terms of AUFSC and best fidelity scores. On the real-world datasets, there is a general trend of scoring worse in comparison to the original experimentation. PBONE is an exception to this trend, getting in close contention with T-GNNExplainer for the best performance on the Wikipedia dataset. On the Reddit dataset, which is the most 'difficult' dataset for the explainers, PBONE outperforms T-GNNExplainer across the board.

The new results broadly reflect better performance for most explainers on both metrics for TGAT trained on both synthetic datasets (v1 and v2). The performance for most explainers on TGN is lower however, except for the baseline explainers on the AUFSC metric.

Observing the results, the salient trend is that T-GNNExplainer scores lower on all metrics in our reproduction of the experimentation compared to the original, except for TGAT trained on the synthetic datasets. Performing T-tests on all new values shows all new values are significantly different, meaning the result from our reproduction is unlikely to be due to chance.

The figures below offer a visual representation of the AUFSC metric. T-GNNExplainer indeed achieves higher fidelity at lower sparsity values (except for TGN trained on Reddit), with PBONE performing second-best.

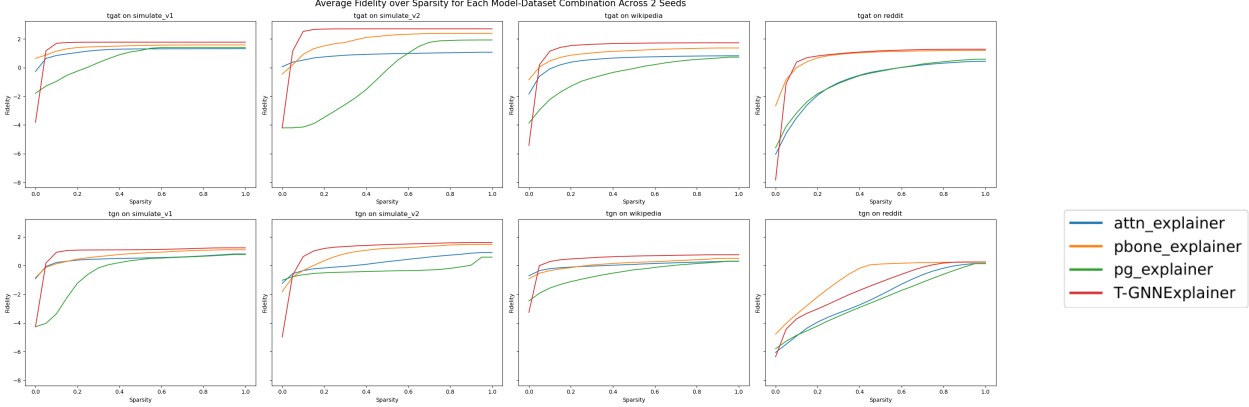

Figure 1: Eight graphs for AUFSC curves and corresponding legend for all four explainers averaged across 2 seeds. The legend indicates the lines for each explainer: blue, orange, green, and red, for ATTN, PBONE, PG Explainer, and T-GNNExplainer respectively.

### 7.1.1 Hyperparameter Tuning

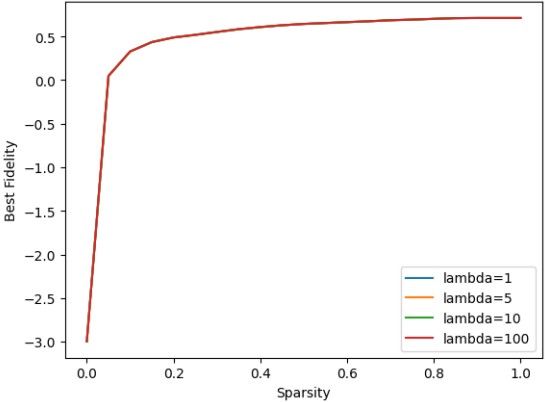

Figure 2: Best fidelity across sparsity intervals on 4 unique lambda values.

We performed tuning on the hyperparameter $\lambda$ to investigate its effect on the explorer in the T-GNNExplainer. A higher value for $\lambda$ encourages more exploration in the node selection step of the explorer (see 3.5). The authors performed this analysis on both models trained on the Wikipedia dataset with $\lambda$ set to 1, 5, 10, and 100. They found that lower values for $\lambda$, which preference exploitation in node selection (particularly $\lambda = 1$), marginally improve performance on the fidelity-sparsity curve. However, they observed that the difference in performance is insignificant.

In our reproduction, values for $\lambda$ set to 5, 10, and 100 all output the exact same results, while $\lambda$ set to 1 output marginally different results that are not noticeable in Figure 2.

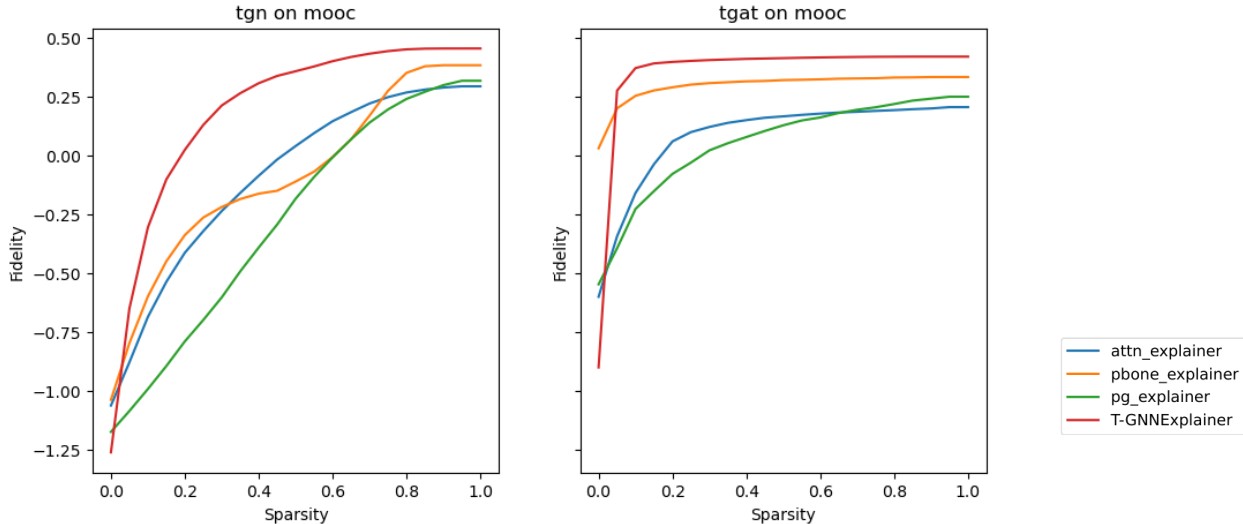

Figure 3: AUFSC curves for both models trained on the Mooc dataset, and corresponding legend for all four explainers. The legend indicates the lines for each explainer: blue, orange, green and red for ATTN, PBONE, PG Explainer, and T-GNNExplainer respectively.

## 7.2 Results beyond original paper

In this section we report our results in Table 7 for the new real-world dataset MOOC. In Figure 3, we illustrate the fidelity-sparsity curve on the new real-world dataset.

Table 7: Best fidelity and AUFSC achieved by each explainer on MOOC, averaged over two seeds.

| | MOOC | | | |
|---|---|---|---|---|
| | **TGAT** | | **TGN** | |
| | MeanBestFid | AUFSC | MeanBestFid | AUFSC |
| **ATTN** | 0.077 | 0.091 | -0.099 | -0.085 |
| **PBONE** | 0.294 | 0.299 | -0.097 | -0.085 |
| **PG** | 0.047 | 0.057 | -0.280 | -0.273 |
| **T-GNN** | **0.339** | **0.369** | **0.172** | **0.201** |

For both models trained on MOOC, T-GNNExplainer outperforms all baseline explainers in both metrics. (See Table 7). PBONE performs competitively with TGAT as the prediction model, however it's inconsistent with TGN as underlying model.

The variability plot in Figure 4 shows a variability in performance of the T-GNNExplainer between the underlying T-GNN models. Using TGAT as an underlying model yields less variance in the outcomes over all models in comparison to TGN.

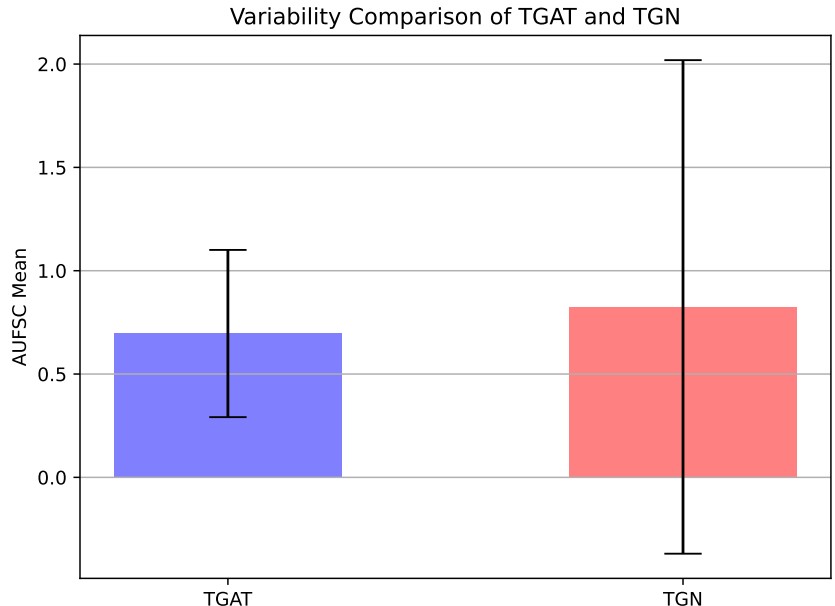

Figure 4: Variability comparison of TGAT and TGN. The mean is taken over all the datasets for the AUFSC metric for each target model, with its standard deviation.

Table 8: The percentage differences between T-GNNExplainer and the best baseline explainer. For example, the first number in the first column indicates that in the original results, T-GNNExplainer outperformed the second-best baseline on the Best Fid metric by 35.6%. This table shows that the difference between T-GNNExplainer and the second best explainer (PBONE) is not as pronounced in our reproduced results as in the original results. For the new results, the negative values indicate where T-GNNExplainer performed second best to PBONE.

| Percentage % difference between T-GNN and best baseline explainer on real-world datasets | | | | | | |
|---|---|---|---|---|---|---|
| **Model** | **Metric** | **Wikipedia** | | **Reddit** | | **MOOC** |
| | | **Old** | **New** | **Old** | **New** | **New** |
| TGAT | Best Fid | 26.64 | 13.91 | 52.85 | 9.63 | 15.68 |
| | AUFSC | 21.19 | 19.61 | 40.50 | 2.68 | 22.23 |
| TGN | Best Fid | 67.89 | 32.02 | 66.83 | 44.43 | 59.44 |
| | AUFSC | 43.41 | 36.95 | 60.19 | 42.78 | 60.28 |

## 8   Discussion

Temporal graph neural networks have the unique ability to infer and predict the developments of evolving network structures (i.e. temporal graphs), in which processes from different scientific disciplines such as ecology and social science (Fortin et al., 2012) (Yu et al., 2018) can be expressed.

In our opinion, an explainer for T-GNNs becomes truly useful when it can successfully leverage the complex and dynamic inhibitory and excitatory influences among events present in real-world processes to provide end-users with computationally feasible and accurate explanations for novel events.

### 8.1 Main claim

Almost all of the values for the AUFSC and Best Fid metrics differ in our reproduction of the original experiment. Despite lower performance in nearly all the output values on both key metrics compared to the original results (See Tables 7, 3, 4, 5, 6), the reproduced experiments still largely support the claim that the T-GNNExplainer outperforms the PGExplainer and ATTN explainers. However, it does not always outperform PBONE. In the instances where T-GNNExplainer does not outperform the baselines, PBONE performs best (Reddit for both TGAT and TGN on both metrics). T-GNNExplainer performs second best in these cases.

In our analysis of the main claim that T-GNNExplainer outperforms the leading baseline by up to $\sim 50\%$ on the AUFSC metric, it is unclear how the original authors calculated this value. We calculated the percentage difference using $\left( \frac{(T-\text{GNN})-\text{best baseline}}{|\text{best baseline}|} * 100 \right)$, and applied it to the original results (See Tables 3, 4, 5, 6). These new calculations differ from the reported percentage difference in the original paper. For example, the original paper states that the T-GNNExplainer outperforms the leading baseline (PGExplainer) for TGN trained on Wikipedia on AUFSC by 86% whereas we found it outperforms by $\sim 700\%$ (See Table 8). Table 8 show that the percentage difference is consistently lower on most comparisons for the new results compared to the old results. This confirms that the the T-GNNExplainer does not perform as well in the reproduced experiments as in the original results, on both the Best Fid and AUFSC metrics.

### 8.2 MOOC dataset

Experiments on the MOOC dataset reveal very similar trends to all explainer models (See Table 7), but the variance of the results is much smaller and closer to 0. The underlying T-GNN models did not score a high accuracy on MOOC compared to the other datasets ( 96% compared to  66% respectively).

Since the prediction of these T-GNN models are used as the "ground truth" for the explainers, the explainer is dependent on the performance of the underlying predictor model. Since the explainer doesn't look "under the hood" at the steps that the T-GNN model takes to make a prediction, and rather uses the T-GNN model to find post-hoc explanations, it follows that when the estimates are noisy, the performance of the explainer suffers.

Considering the low accuracy of both TGN and TGAT on MOOC, this could explain why the variance of the results on MOOC is far smaller than on other datasets.

### 8.3 Model agnostic claim

Although the authors claim that T-GNNExplainer is model agnostic, it doesn't seem to stay true for these two SOTA models. Our results show that the explanatory power of T-GNNExplainer differs between the TGAT and TGN models. Figure 1 shows that the T-GNNExplainer performs worse overall on the TGN model than TGAT, and and Figure 4 shows performance on the TGN model has higher variance, so it also offers less certainty.

### 8.4 Reddit dataset

All explainers underperform on the Reddit dataset. One possible explanation points to the size of the dataset and the method for splitting the data. Reddit is by far the biggest dataset ($\sim 260000$ bigger than MOOC). When training the T-GNN models, the data is split temporally where the first 85% events are used for training and validation and the last 15% events are used as test data. The problem with this method is that graph data is not independent, so events earlier in time could influence events later in time. Separating the data in this way can lead long-distance predictions that are inherently more uncertain. In line with this is MOOC, which is the second biggest dataset, also resulting in low scores comparatively. More advanced temporal splitting methods could be used to obtain more valid outcomes (de Bruin et al., 2021).

Another reason could be that the Reddit dataset seems to have mistakes in it. It is said to be a bipartite dataset representing users editing pages. However there are nodes that appear in both sets, which is impossible in a bipartite dataset. This could result in worse performance as the data is not clean.

## 8.5 Hyperparameter Tuning

The negligible difference in performance that is measured in all intervals of the hyperparameter $\lambda$ suggests that including the exploration term in the node selection process of the explorer is largely insignificant in comparison to the exploitation term, because the pre-trained navigator is very dominant in finding optimal subgraphs, in comparison to exploring novel options.

Given that the Monte Carlo Tree Search in the T-GNNExplorer is one of the most computationally slow and costly procedures of the model, the insigificance of the exploration term raise questions whether the explorer could benefit from other, more efficient methods of subgraph-search.

## 8.6 Computational efficiency

The authors discuss the computational efficiency of the T-GNNExplainer in comparison to the baseline models in their Appendix. They found that the T-GNNExplainer is significantly slower than the baseline models, none of which are search-based methods. For example, T-GNNExplainer takes 28.2s to explain one instance in the Reddit datset trained on TGAT compared to 0.39s for PBONE (Appendix by (Xia et al., 2023)). They state that the complexity of the MCTS algorithm is O(NDC) (Appendix by (Xia et al., 2023)), where N is the number of rollouts, D is how far each rollout expands a node, and a constant for inference time. Given that in the code, the hard coded leaf nodes meant that each rollout expanded nodes all the way down to a subgraph containing 1 event, the complexity was very high. Efficiency would likely be improved if this rollout search wasn't continued to the final node.

Given the extreme computational demand detailed in Section 6.3 for merely marginal improvements over simpler explainers such as PBONE, it is unclear whether the use of T-GNNExplainer can be justified despite its superior performance. Beyond performance, accessibility is an important factor in measuring the "usefulness" of tools such as T-GNNExplainer, especially for teams with limited funds, time, or resources.

## 8.7 Discrepancy between results

Given that seeding was not properly implemented in the original study, it makes sense that none of the values in the reproduced experiments match the original results, however it is unclear why the values differ so drastically. One interesting discrepancy is that in our results, PBONE either performs second best, or in some cases, outperforms T-GNNExplainer. In the original results, the leading baselines are either ATTN or PGExplainer. PBONE performs best on AUFSC for both models trained on the Reddit dataset. In instances where PBONE performs second best, its evaluation metrics are often close in value to those of T-GNNExplainer.

## 8.8 Conclusion

Since T-GNNExplainer is computationally intensive given the task (i.e. it only provides instance level explanations), it is not clear whether it is model-agnostic, and it requires very precise dataset configurations, it would be valuable to explore other methods to provide explanations for T-GNNs. In comparison to T-GNNExplainer, PBONE is a very simple model that is computationally much faster than T-GNNExplainer. Considering its simplicity, computational efficiency, and its ability to "compete" with T-GNNExplainer according to our findings, refining perturbation-based methods could be a valid avenue for exploration. Justification could be found in similar methods explored by Lucic et al. (2022) where the authors use a perturbation-based explained method on GNNs to achieve strong results.

## 9   What was easy and what was difficult?

**What was easy:** The authors made their code accessible online, without which the T-GNNExplainer would not have been reproducible based on the paper alone. Reading the responses on OpenReviews was helpful in providing insight into the difficulties others also had with the paper. The Appendix provided by the authors in response to the OpenReview comments helped clarify questions we had about hyperparameters and implementation.

**What was difficult:** Discrepancies between the code and the paper, and inconsistencies in the code (missing directories, dependencies, hard-coded hyperparameters, inconsistent nomenclature, improper seeding and GPU allocation, etc.) meant the code required a lot of trial-and-error to produce valid results. For example, the $\lambda$ variable named "c_puct" was reported to be 5 in the paper, but was set to 100 in the config files. This discrepancy was not identified until after the T-GNNExplainer had already been trained. Similarly, the original code was not seeded properly so all experiments had to be re-trained again. This trial-and-error, combined with the massive compute requirements demanded by the code, meant we had to forego many crucial experiments due to lack of computational resources and high associated costs. Using the Machine Learning Emissions Calculator (Lacoste et al., 2019), we estimate that the GPU resources on this project alone emitted at least 103 kg $CO_2$ eq., or equivalent to driving more than 400km by an average ICE car (See Table 9 in Appendix A).

While validating reproducibility is essential in machine learning, there is a serious conversation to be had about the trade-off between these efforts and the societal and environmental costs incurred when the groundwork to enable reproduction is not "baked" into the research process from the start.

**Communication with original authors:** The authors were contacted, but we did not receive a response.

## Acknowledgements

This research was supported by the Google Cloud Research Credits program with the award 326120695. We'd also like to thank Martin Smit for his close guidance and Florian Golemo for his invaluable advice and support in running the experiments. This work was created in the context of the course "Fairness, Accountability, Confidentiality and Transparency in AI" at the Universiteit van Amsterdam (UvA) and we would also like to thank the lecturer, Prof. Fernando Pascal Dos Santos.

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

## A  Machine Learning Emissions Calculator

In Table 9, we provide the specifications for the calculator used to calculate the $CO_2$ impact.

Table 9: Specifications for the calculator (Lacoste et al., 2019)

| Hardware | Hours used | Provider | Carbon Efficiency | Offset Bought |
|----------|-----------|----------|-------------------|---------------|
| AMD EPYC 7763 | 400 | Private Infrastructure | 0.459 | 0 |
| T4 | 2760 | Google Cloud Platform | - | 0 |

## B  Experimental setup: Novel datasets

Table 10: Hyperparameters used for all models on all datasets.

| Hyperparameter | TGAT | | | TGN | | |
|----------------|------------|-----------|------|------------|-----------|------|
| | Real-world | Synthetic | MooC | Real-world | Synthetic | MooC |
| Hidden Dimension | 172 | 4 | 4 | 172 | 4 | 4 |
| Attention heads | 2 | 2 | 2 | 2 | 2 | 2 |
| N degree | 10 | 10 | 10 | 10 | 10 | 10 |
| Memory dimension | - | - | - | 172 | 4 | 4 |
| Time dimension | 172 | 4 | 4 | 172 | 4 | 4 |
| Node feature dimension | 172 | 4 | 4 | 172 | 4 | 4 |
| Edge feature dimension | 172 | 4 | 4 | 172 | 4 | 4 |
| Training epoch | 10 | 100 | 100 | 10 | 100 | 100 |
| Learning rate | $1e^{-4}$ | $1e^{-4}$ | $1e^{-4}$ | $1e^{-4}$ | $1e^{-4}$ | $1e^{-4}$ |
| Batch size | 512 | 256 | 512 | 256 | 512 | 256 |
| Dropout probability | 0.1 | 0.1 | 0.1 | 0.1 | 0.1 | 0.1 |

The format of the MooC dataset matches the required format of real world datasets. Similar to Reddit and Wikipedia, feature normalization is applied to the dataset. For the Reddit hyperlinks dataset, the concrete preprocessing steps that are taken can be found in the Reddit_hyperlinks_preprocess.ipynb file. In summary, min-max normalization is applied to each feature column individually and the datatypes and column names are changed to match the requirements set by the author's code. Once these two steps are completed the instructions in the Read.me file provided can be used to train to T-GNNExplainer on a novel dataset.

The process.py file ensures the non-feature columns exclusively contain two node indexes, a timestamp and a label. After processing the file is saved in a subdirectory called 'processed'. Secondly, indexes ought to be generated for a real-world dataset. This is done by calling 'python tg_dataset.py -d dataset_name -c index'. The read.me file gives the wrong instruction here. This step randomly samples 500 events from the test split on which the performance of the T-GNNExplainer will be evaluated. Before T-GNNExplainer can be trained, the GNN's (TGAT and TGN) need to be trained on the new dataset. Make sure that the hyperparameters are correctly specified in the run.sh files. For the inductive event prediction performed by the two GNNs, average precision after each epoch is outputted in the terminal. In our experimentation, the best achieved average precision is recorded (Table 11).

Once TGAT and TGN are trained on all datasets. T-GNNExplainer is ready to be trained. The final step of the Read.me provides adequate instructions. The Fidelity results are outputted in files called model_name_lownumber_to_highnumber.csv.

Table 11: Models' average precision (AP) for the inductive event prediction on all datasets.

|        |     | Wikipedia | Reddit | Synthetic v1 | Synthetic v2 | MooC  |
|--------|-----|-----------|--------|--------------|--------------|-------|
| **TGAT** | New | 0.971 | 0.982 | 0.945 | 0.953 | 0.626 |
|          | Old | 0.979 | 0.975 | 0.963 | 0.964 | 0.706 |
| **TGN**  | New | 0.947 | 0.980 | 0.945 | 0.960 | 0.659 |
|          | Old | 0.985 | 0.966 | 0.954 | 0.969 | 0.712 |

