# OpenReview forum: "[Re] Reproducibility Study of “Explaining Temporal Graph Models Through an Explorer-Navigator Framework""
_TMLR — Accepted by TMLR_

### Review · Reviewer_qRyC · 2024-03-11

**Summary Of Contributions:**

This paper presents a reproducibility report of the paper "Explaining Temporal Graph Models Through an Explorer-Navigator Framework". The authors re-run the experiments for baselines and also TGNN-Explainer, and also propose a new dataset MOOC for evaluation. The results obtained with the reproducibility report is that TGNN-Explainer does not outperform baselines as much as it claims to do.

**Audience:**

Yes

**Broader Impact Concerns:**

No.

**Claims And Evidence:**

No

**Requested Changes:**

See the "Weaknesses".

**Strengths And Weaknesses:**

## Strengths
1. Reproducibility is important in machine learning, and this paper, (I don't know whether it should be 'fortunately' or 'unfortunately') identifies a paper that is not 'so' reproducible.
2. The authors identify the necessary steps needed to run the code of the paper, which is even missing from the original authors. I appreciate the efforts it takes to fill the missing steps.
3. The MOOC dataset is of different properties as Reddit and Wikipedia (the original datasets), and indeed, results on MOOC is quite different from those obtained with wikipedia.

## Weaknesses
1. I would like to first question the impact of whether the reproducibility report would make an impact. As of March 2024 when I review this paper, the studied "T-GNNExplainer" only has 5 citations in about one year. This means that the T-GNNExplainer itself has rather limited impact, and I am not sure whether reproducing this would raise the interests of related researchers. Of course there will be 'some' researchers who are interested, but I am not sure whether it is sufficient to motivate the paper.
2. The paper's organization has to be improved, especially within the section of experimental results. For example, the authors could have shown complete Tables 2 and 3 with both datasets and both models, but they did not, which make reading the experimental results quite difficult.
3. I think that the hyperparameter tuning is necessary as an ML model should be tested on multiple hyperparameters to determine its performance, but the authors did not do so.
4. Miscellaneous:
     - There are broken references '??'
     - Section 3.2.1 can be better written by giving some intuitions on the Equations.
     - Figure 1 can be better plotted with clearer legends to separate different models.
     - It seems that PBONE performs very differently between 'old' and 'new'. Any reasons why this may happen?

---

> ### Author Response · Authors · 2024-04-22
> **Response to Reviewer qRyC**
>
> We'd like to thank all reviewers for their time, effort, and the great comments that helped dramatically improve our paper. Below we have addressed the key weaknesses you identified in our paper:
>
> Weakness 1: This field is difficult and underexplored. Temporal graphs are influential data structures on which successful explanation can be very valuable. Our team decided to pick this research paper as it stood out to us with technical methodology and its strong claims of explainer performance. In our vision, reproducibility work should not be limited to the collection of papers that receive the most academic traction.
>
> Our results and experience in attempting to reproduce this paper directly highlight the reproducibility crisis in machine learning and the dire need for competitions such as TMLR. The code was difficult to run, computationally demanding and expensive, and not properly seeded; our process and findings are important to share with the machine learning community as a lesson in what is needed to do good, reproducible research.
>
> Weakness 2: We proofread the document for writing clarity, typos, and structural organization.
>
> Weakness 3: We reran the experiments on two seeds to ensure robust results, however due to the extreme computational demand, the hyperparameter tuning experiments could not be completed in time for the re-submission deadline. We agree that hyperparameter tuning is important and once the experiments are complete, results will be updated in the paper.
>
> We retrained all models (TGN and TGAT) on two seeds for all datasets. Additional code refactoring was required to make the code run with robust seed implementation. Seeds were not correctly implemented in the original code so directly reproducing the exact original results is not possible. We updated the section on computational requirements to reflect the large compute demand of these additional experiments.
>
> We are currently running the additional experiments that the reviewers requested. So far with 1000€ in compute costs expended over the last two weeks alone, we managed to finish 70% of the experiments (two seeds on all datasets and all models with all explainers except the TGNNExplainer). If the paper is accepted the TGNNExplainer results will be included. A third seed was not possible due to time restrictions but would be included in the camera-ready submission.
>
> Weakness 4: See weakness 2 re: typos and clarity. Discussion on the performance of PBONE can be found in the discussion section 8 “Discrepancy between results”.
>
> Final thoughts:
> - Despite not having the results for TGNNExplainer, the fidelity-sparsity graphs averaged over two seeds for the other baselines on the real datasets largely hold with our initial reproduction and the original paper. The simulated datasets however reveal some variance in the results, which raise questions regarding whether these datasets are valid to use in these experiments.
> - Once the final results are ready, the paper will be updated with the above changes and updated values and charts.

---

### Review · Reviewer_H3zU · 2024-04-05

**Summary Of Contributions:**

This paper replicates the findings of Xia et al. (2023) for the temporal GNN explainer. TGNNExplainer is a navigator neural network that learns to detect subgraphs that are sufficient to classify correctly a particular event in the dataset i.e. explain sufficiently the predicted event. The authors verify the original tests, ensure their clear reproducibility and compare with the same baseline methods, in order to affirm T-GNNExplainer's superior performance in the predefined metrics. The study also reveals some degree of variability in performance across different models and datasets and some incosistencies with the original paper results. This uncovers the need for more elaborate experiments and consideration of alternative methods.

**Audience:**

Yes

**Broader Impact Concerns:**

There is no obvious ethical concern regarding the imapct of the study.

**Claims And Evidence:**

Yes

**Requested Changes:**

The experiments need to be validated across many different seeds to quantify the robustness of the method.

To increase the scope of the study sufficiently, the authors could include more real world datasets varying in size and time steps. Moreover, as the author suggest in future work, it is meaningful to examine the effectiveness in node and graph tasks, and not only on bipartites tasks. The methods are also limited as more GNN architectures like GCN, SAGE and GIN should be tested to increase the scope of the analysis.

Finally, the paper needs proofreading e.g. There is a table captions called “Percentage % difference between best and best baseline explainer” also, some typos e.g. Figure ??, Figure 2 ??

**Strengths And Weaknesses:**

Strengths:

Graph explainers are a timely topic with many important applications from biology and chemistry to e-commerce and weather forecasting.

The authors perform a rigorous analysis of the referred paper, examining each claim separately and go to notable extend to make their work easily reproducible and accessible.

A new dataset is tested and the new results suggest a drop of the performance, although the overall ranking is retained.

A pattern is uncovered regarding the relationship between the number of nodes of the “explainer” subgraph and the performance i.e. the more dense the better the metric.

Weaknesses:

The experiments are limited, in the sense that error bars are missing.

The study needs to be more broad to constitute a contribution.

The authors could elaborate more on the uncovered patterns, such as the one regarding the density of the subgraph and the explainability e.g. ablation studies to further butress the observation and potentialy test the potential of the pattern to improve the results.

---

> ### Author Response · Authors · 2024-04-22
> **Response to Reviewer H3zU**
>
> We'd like to thank all reviewers for their time, effort, and the great comments that helped dramatically improve our paper. Below we have addressed the key changes you requested in our paper:
>
> Request 1: We retrained all models (TGN and TGAT) on two seeds for all datasets. Additional code refactoring was required to make the code run with robust seed implementation. Seeds were not correctly implemented in the original code so directly reproducing the exact original results is not possible. We updated the section on computational requirements to reflect the large compute demand of these additional experiments.
>
> We are currently running the additional experiments that the reviewers requested. So far with 1000€ in compute costs expended over the last two weeks alone, we managed to finish 70% of the experiments (two seeds on all datasets and all models with all explainers except the TGNNExplainer). If the paper is accepted the TGNNExplainer results will be included. A third seed was not possible due to time restrictions but would be included in the camera-ready submission.
>
> Request 2: We acknowledge the value of different base models such as GCN, SAGE to be tested to increase the scope of the analysis, however including these is computationally not feasible in this time span unfortunately.
>
> Request 3: We proofread the document for writing clarity & typos.
>
> Final thoughts:
> - Despite not having the results for TGNNExplainer, the fidelity-sparsity graphs averaged over two seeds for the other baselines on the real datasets largely hold with our initial reproduction and the original paper. The simulated datasets however reveal some variance in the results, which raise questions regarding whether these datasets are valid to use in these experiments.
> - Our results and experience in attempting to reproduce this paper directly highlight the reproducibility crisis in machine learning and the dire need for competitions such as TMLR. The code was difficult to run, computationally demanding and expensive, and not properly seeded; our process and findings are important to share with the machine learning community as a lesson in what is needed to do good, reproducible research.
> - Once the final results are ready, the paper will be updated with the above changes and updated values and charts.

---

### Review · Reviewer_JK87 · 2024-04-08

**Summary Of Contributions:**

The authors performed a reproducibility study of "Explaining Temporal Graph Models Through an Explorer-Navigator Framework" focusing on a subset of the main claims, specifically the claim of TGNNE outperforming baselines by up to 50% and agnosticism w.r.t to the underlying predictor.

For this they reproduced the original paper using the papers supplementary, disambiguating paper/code divergences in favour of the paper.
They find varying behaviour on multiple levels, first receiving markedly different final accuracies and evaluation metrics on the original datasets, second finding that the TGNNE does not outperform all baselines on all datasets, third finding that there is a dependency on the underlying predictor models and datasets.
The authors also extend experimentation by attempting to train and evaluate on the MOOC dataset and perform an analysis of the dependence on dataset structures, claiming a dependence on subgraph density.

Finally, challenges encountered in the reproducibility study are remarked upon, in particular the difficulty of reproducing papers not released with reproducibility "built in"

**Audience:**

Yes

**Broader Impact Concerns:**

I think the broader impact concerns are appropriate

**Claims And Evidence:**

No

**Requested Changes:**

Critical: Please perform suitable statistical signficance tests, adding training seeds and evaluations until the results are meaningful *OR* strongly disclose the limitations implied by not doing so. I acknowledge the difficulty of compute constraints, but would request *at least* adding one or two additional runs to establish a certain minimum of variability investigation + a disclosure that more is required if compute is the limitation (this can then further be discussed in the paper).

Doing at least one of these is critical, otherwise I cannot claim that this paper is supported in its claims by evidence, much as I appreciate reprodicibility work.

Strengthening:
Aside from this, I would adjust the reporting of differences and clean up the writing/typo nitpicks I voiced.

**Strengths And Weaknesses:**

---

Strengths:

+++ authors appear to have conducted a thorough reproduction, in partilar disambiguating hyperparameters and overcoming other common obstactles to reproducibility

++ upon finding divergence, further investigation was performed in order to attempt to explain divergence, including adding a new dataset

\+ presentation is at times idosyncratic (see below) but clear and understandable.


Weaknesses:
(---, maybe?) unsure whether this counts as (self?)-plagiarism, but the appendix in the supplementary appears to be 1:1 the supplementary of the underlying paper

--- while I understand and empathize with the computational constraints, no mention is made of multiple seeds (i.e., trianing multiple models) nor is there a systematic evaluation of possible randomness in the results - since the original paper does not mention training multiple seeds per model either, this is understandable, but it leaves occams razor for the failed reproduction (variability across seeds) unexplored.

--- likewise, in order to be meaningful, some of the observations really need statistical significance test, in particular the regression fit in fig 2a and the results reported in tables (the latter also needs evaluation of variability).

while I want to state that these are weaknesses inherited from the original paper, they remain hindrances for me to full recomennd this paper, especially without explicit acknowledgement.
I think the compute budget should have gone to those additional seeds and significance tests first, not on attempting to add more datasets.

-- why was the percentage difference computed in a way to make "it not possible to compute percentage differences" of metrics on page 10/11? what hinders us from normalizing the values to 0-1 and then computing scores, or just from properly defining the percentage calculation to ensure sensible results?

\- at some points the writing can be a bit unclear. E.g., in 8.2, "did not score a high accuracy on MOOC compared to the other datasets" is followed by (96% compared to 66% respectively, I presume 66% refers to mooc and 96% to the others, but the flip is confusing)

(-) nitpick: some typos/leftover from wanting to add two datasets: page 11 bottom, page 12 top and some other places refer to "datasets" where it should be singular, page 12 top also says "it would fruitful" where it should read "it would *be* fruitful". Also, broken reference to fig 2b in 8.4


(-)  earlier in the paper the authors refer to "connections in both directions"; in 8.5. paragraph 2, you correctly refer to "both sets"; is this another small mistake?

---

> ### Author Response · Authors · 2024-04-22
> **Response to Reviewer JK87**
>
> Dear reviewer,
>
> We'd like to thank all reviewers for their time, effort, and feedback that helped dramatically improve our paper. Below we have addressed the key weaknesses you identified in our paper:
>
> Weakness 1: The unabridged description of the methodology was originally moved to the appendix to shorten the length of the paper, but we found it valuable to provide the full description in the appendix to make the explanation from the original paper more accessible. Given that our paper is already considered a long submission, we moved the detailed methodology from the appendix back into the paper.
>
> Weakness 2: We retrained all models (TGN and TGAT) on two seeds for all datasets. You are correct in observing that this was a weakness inherited from the original paper - additional code refactoring was required to make the code run with robust seed implementation. Seeds were not correctly implemented in the original code so directly reproducing the exact original results is not possible. We updated the section on computational requirements to reflect the large compute demand of these additional experiments.
>
> We are currently running the additional experiments that the reviewers requested. So far with 1000€ in compute costs expended over the last two weeks alone, we managed to finish 70% of the experiments (two seeds on all datasets and all models with all explainers except the TGNNExplainer). If the paper is accepted the TGNNExplainer results will be included. A third seed was not possible due to time restrictions but would be included in the camera-ready submission.
>
> Weakness 3: Once results are finalized, statistical significance tests will be performed.
>
> Weakness 4: Values have been normalized to compute scores.
>
> Weakness 5 & 6: We proofread the document for writing clarity.
>
> Final thoughts:
> - Despite not having the results for TGNNExplainer, the fidelity-sparsity graphs averaged over two seeds for the other baselines on the real datasets largely hold with our initial reproduction and the original paper. The simulated datasets however reveal some variance in the results, which raise questions regarding whether these datasets are valid to use in these experiments.
> - Our results and experience in attempting to reproduce this paper directly highlight the reproducibility crisis in machine learning and the dire need for competitions such as TMLR. The code was difficult to run, computationally demanding and expensive, and not properly seeded; our process and findings are important to share with the machine learning community as a lesson in what is needed to do good, reproducible research.
> - Once the final results are ready, the paper will be updated with the above changes and updated values and charts.

---

### Author Response · Authors · 2024-03-05
**Question about decision status**

Dear TMLR staff,

We expanded our set of recommended reviewers from 2 to 6 on 26 February 2024. The minimum requirement of at least 3 recommendations was met 1 day after the recommendation deadline. Our recommendation selection became incomplete because Sinead A. Williamson was removed as a reviewer after we selected her as a recommendation at the date of original submission. This is the reason the number of recommendations for our paper dropped to lower than three at the recommendation deadline of 25 February. We hope that our paper is still subject to evaluation and not disqualified because of this unfortunate course of events. An update on the evaluation status would be greatly appreciated.

---

### Decision · Action_Editor_8WuN · 2024-05-10

**Recommendation:** Accept as is

**Comment:**

The manuscript delivers a reproducibility study pertaining to the paper “Explaining Temporal Graph Models Through an Explorer-Navigator Framework” (Xia et al., 2023).  Reviewers generally appreciated the efforts made for conducting the study, as well as the solidity of its findings.  There were some concerns with regards to statistical significance and clarity of writing, but the authors largely addressed those.  A point of disagreement within the committee relates to the limited significance and novelty of the work, and in particular to whether these limitations should be an impediment for publication.  While I share the sentiment that they may constrain the manuscript's future impact, TMLR's acceptance criteria deliberately avoids rejection on the grounds of such sentiments (especially for reproducibility studies).  I am therefore recommending acceptance, leaving it to TMLR readership to determine the impact of the contribution.

**Audience:**

Although the group of potentially interested readers may not be large, there will definitely be some individuals in TMLR's audience that will likely be interested in the manuscript.

**Claims And Evidence:**

Claims in the manuscript are properly supported by evidence.